# Genomic Analysis of *Yersinia pestis* Strains from Brazil: Search for Virulence Factors and Association with Epidemiological Data

**DOI:** 10.3390/pathogens12080991

**Published:** 2023-07-28

**Authors:** João Luiz de Lemos Padilha Pitta, Matheus Filgueira Bezerra, Diego Leandro Reis da Silva Fernandes, Tessa de Block, Ane de Souza Novaes, Alzira Maria Paiva de Almeida, Antonio Mauro Rezende

**Affiliations:** 1Microbiology Department of Aggeu Magalhães Institute—FIOCRUZ PE, Recife 50740-465, PE, Brazil; matheus.bezerra@fiocruz.br (M.F.B.); leanreis7@gmail.com (D.L.R.d.S.F.); alzira.almeida@fiocruz.br (A.M.P.d.A.); 2Bioinformatics Platform of Aggeu Magalhães Institute—FIOCRUZ PE, Recife 50740-465, PE, Brazil; 3Department of Clinical Sciences—Institute of Tropical Medicine, 2000 Antwerp, Belgium; tdeblock@itg.be; 4Department of Biological Sciences—Federal University of Vale do São Francisco—UNIVASF, Petrolina 56300-000, PE, Brazil; ane.snovaes@gmail.com

**Keywords:** *Yersinia pestis*, plague, virulence factor, comparative genomics

## Abstract

*Yersinia pestis*, the etiological agent of the plague, is considered a genetically homogeneous species. Brazil is currently in a period of epidemiological silence but plague antibodies are still detected in sentinel animals, suggesting disease activity in the sylvatic cycle. The present study deployed an in silico approach to analyze virulence factors among 407 Brazilian genomes of *Y. pestis* belonging to the Fiocruz Collection (1966–1997). The pangenome analysis associated several known virulence factors of *Y. pestis* in clades according to the presence or absence of genes. Four main strain clades (C, E, G, and H) exhibited the absence of various virulence genes. Notably, clade G displayed the highest number of absent genes, while clade E showed a significant absence of genes related to the T6SS secretion system and clade H predominantly demonstrated the absence of plasmid-related genes. These results suggest attenuation of virulence in these strains over time. The cgMLST analysis associated genomic and epidemiological data highlighting evolutionary patterns related to the isolation years and outbreaks of *Y. pestis* in Brazil. Thus, the results contribute to the understanding of the genetic diversity and virulence within *Y. pestis* and the potential for utilizing genomic data in epidemiological investigations.

## 1. Introduction

The plague is an infectious disease caused by *Yersinia pestis*, a nonmotile, aerobic Gram-negative bacillus that has already devastated human populations throughout its three major recorded pandemics [1]. The first pandemic, called the Plague of Justinian, devastated the civilized world between 542 and 602 AD. The second plague pandemic started in Asia and reached Europe and North Africa, enduring from the XIV until the XVI century. It claimed the lives of one third of the European population in the period from 1346 to 1353, which, nowadays, is commonly known as the Black Death [2]. Plague was introduced into Brazil during the current pandemic, in 1899, by rats and fleas present in commercial steamships in the port of Santos in the State of São Paulo [3]. Due to the transport and trade of goods, the plague spread inland through roads, railways, and river transportation, reaching, initially, countryside villages. In some regions, the plague could establish new foci by maintaining a sustainable cycle in sylvatic rodents from local ecosystems, causing many outbreaks in rural landscapes within the Caatinga biome. These outbreaks occurred across several states in the Northeast region (Ceará, Rio Grande do Norte, Paraíba, Pernambuco, Alagoas, Bahia, and Piauí) and the northeast of Minas Gerais (Vale do Jequitinhonha region), as shown in Figure 1. Moreover, outside the Caatinga, there is a smaller plague transmission area in the Atlantic Forest biome: Vale do Rio Doce in Minas Gerais and Serra dos Órgãos in the state of Rio de Janeiro [3,4,5]. The distribution of *Y. pestis* cases in Brazil, according to the National Reference Service in Plague reported from 1966 to 1997 (data not published), is represented in Figure 1.

The belief that the plague has already been eradicated and that it is restricted to the distant past of humanity is a serious mistake, as it continues to be active and causes deaths throughout the world. The most recent data shows records of infection in several parts of the world, especially in Madagascar, and the African continent is the place with the highest incidence of cases. Only in Madagascar, 2323 cases were recorded between 2013 and 2018, and there are also records of 410 cases in the Democratic Republic of Congo, 36 cases in Tanzania, and 22 in Uganda [6,7]. In the same period, some cases were reported in Asia, China, and Mongolia, with five cases reported in each of these countries. In addition, in South America, 40 cases were registered in Peru and 3 in Bolivia. Even in more developed countries, with greater control and better sanitary conditions, cases have been registered, such as the 40 cases registered in the United States between 2013 and 2018 [6,7]. This large number of cases that continue to occur around the world, along with the great virulence of *Y. pestis,* are associated with a refined arsenal of gene regulation. The complex life cycle of the bacillus includes the ability to grow and multiply in diverse organisms, such as ectoparasites and mammals, and the regulation of virulence factors that are decisive to host adherence, modulation of the immune response, interruption of cellular metabolism and to obtain essential nutrients [8]. From a genomic point of view, *Y. pestis* is widely known to be highly conserved, especially when comparing the second great pandemic strains with modern strains [1]. Observing the reference strain KIM (GCF_000006645), which originated from the second pandemic, and the modern reference strain CO92 (GCF_000009065), they both have a genome of 4.6 Mb, just over 4000 ORFs and CG content around 47%, also sharing major virulence genes and more than 95% of the genome [9,10]. The strains associated with the first pandemic differ from the modern ones, with the absence of genes such as *pheA* and *celB*. Strains from the final period of the first pandemic also have lost *mgtB* and *mgtC* genes, a fact that was associated with the end of the first pandemic [1,11]. Throughout history, the evolution of *Y. pestis* has had two notable points. First, it diverged into an ancient strain known as the Neolithic Gok2 approximately 5700 years ago, established as an independent lineage in Eurasia [12]. Secondly, during a timeframe spanning from 3300 B.P. to 5300 years ago, it diverged into different strains that acquired all the necessary genetic components for transmission through fleas. This included the integration of the *ymt locus*, which enabled the bacillus to survive within the midgut/proventriculus of fleas and inside macrophages [13,14,15].

## 2. Materials and Methods

### 2.1. Biological Data

The present study used 411 genomes of *Y. pestis* belonging to the National Reference Service in Plague (Serviço de Referência em Peste—SRP). The strains were isolated from rodent, flea, and human cases from epidemic and endemic periods during surveillance activities since 1966 and were maintained in plain agar stubs, in rubber-stoppered tubes, under refrigeration at 4 °C [16]. These genomes are deposited in the NCBI database under BioProject: PRJNA421720. Genomes were sequenced using the Illumina HiSeq 2000 platform (Illumina, Inc., San Diego, CA, USA) using standard protocols for the TruSeq SBS Kit v3-HS (Illumina, Code FC-401-3001) [3]. The strains belong to the *Y. pestis* cultures collection Fiocruz/CYP (http://cyp.fiocruz.br/accessed on 4 April 2019) of the Aggeu Magalhães Institute (IAM/Fiocruz-PE) [17]. The collection began as an initiative of the Brazilian government and the World Health Organization (WHO) in which a broad research program was developed for plague control in the Araripe plateau between 1966 and 1974. In 1974, the collection was transferred to the central laboratory of Garanhuns, which operated as a plague diagnosis center for the entire country of Brazil until 1982 and, finally, it was transferred to the Aggeu Magalhães Institute, receiving new strains until 1997, when the last strains were isolated. This collection is part of SpeciesLink, a distributed information system that integrates primary data from scientific collections and is registered with the World Federation Culture Collection (WFCC).

### 2.2. Evaluation of Genome Sequencing

The quality of the sequencing data of the 411 strains of *Y. pestis* was evaluated using FastQC 0.11.8 software [18]. FastQC generated data such as the total number of sequences, sequence sizes, percentage of GC content, and the Phred (quality measurement value) of the bases. Due to the large number of sequences and data generated by FastQC, MultiQC v1.10 [19] was used to consolidate the results that have an average 2.2 million of reads, GC content of 47%, and average depth of 42×. Sequencing data numbers are available in Appendix A.

### 2.3. Genome Assembly and Annotation

Sequencing data was filtered using Trimmomatic v. 0.38 [20] using the values of Phred 33 and the parameters LEADING:10, AVGQUAL:30, TRAILING:10, SLIDINGWINDOW:4:15, and MINLEN:60. For each isolate, the paired-end filtered data were used in de novo assembly of the genomes. The assembly was performed using the VelvetOptimiser script, developed to optimize parameters for reassembly using the Velvet program [21], which was configured with an N50 optimization function for k-mer selection. The analysis of the assembled genomes was performed using GenomeTools 1.5.8 [22] and the results were aggregated using an in-house script. The genome annotation process was performed with the Prokka pipeline [23], which includes several tools such as Prodigal [24] and SignalP [25] to annotate relevant genome features within contigs, such as coding sequences (CDS) and signal leader peptides.

After assembly, 5 strains (Yp3290, Yp3202, Yp3296, Yp3241, and Yp3959) were excluded from the study, resulting in 407 genomes for the subsequent analyses. Excluded strains had genomes sizes far below the average (4.6 Mb) expected for *Y. pestis* and the number of reads was also lower compared with other strains and could compromise the analyses. Data regarding the assembly, their respective sizes, and the number of reads, including removed strains, can be found in Appendix A.

### 2.4. Pangenome Analysis

The data from the functional annotation performed by Prokka was used as input to the pangenomics analysis using the Roary software V.3.11.2 [26]. Pangenomics analysis allowed the description of genes shared by several strains (core genome) [26]. This analysis generated a matrix of the presence and absence of accessory genes and a phylogenetic tree based on this matrix. To generate a visualization of the standard Roary results, the roary_plot.py script was used. This script uses Roary’s output files to produce a pie chart showing the distribution between core and accessory genes and the phylogenetic tree with the matrix of presence and absence of genes. The data relating to the strains and their associated clades can be found in Appendix A.

### 2.5. Statistical Analysis of Presence and Absence Genes in Pangenomics

Statistical analysis of the presence and absence of genes for the 407 strains was performed with Scoary V.1.6.16 [27] using Fisher’s exact test, which is a statistical significance test model for analyzing contingency tables with hypergeometric distribution. Scoary crossed the data from Roary pangenomic analysis against phenotypes treated as binary data from strains belonging to the clades highlighted in the Roary phylogenetic tree. This analysis allowed for highlighting present or absent genes that are specific to these clades, grouped by statistical relevance, in addition to bringing numbers on how many isolates a gene has within a clade, how many do not have a gene within a clade, and how many have the gene in other strains that do not belong to a clade. All of Scoary’s analysis results are available in Appendix A.

### 2.6. Identification of Virulence Factors

To identify virulence genes in the assembled genomes of the 407 strains of *Y. pestis*, ABRicate V1.0.0 was used (https://github.com/tseemann/abricate accessed on 26 July 2021) with the Virulence Factor Database (VFDB) [28]. ABRicate allowed for the automatization of the analysis given the data volume. The downloaded VFDB database was used as a subject set to perform a local alignment in the predicted genome’s CDS with the BLASTN 2.10.1+ algorithm [29]. Since some proteins with statistical relevance in the clades were previously annotated as hypothetical, we further investigated this group of genes using InterProScan V 5.56-89.0 [30], which employs approaches such as the Hidden Markov Model (HMM) or local alignment and utilizes predictive models, known as signatures, from multiple databases to conduct functional analysis of proteins. It classifies proteins into families, predicts domains, and identifies important functional elements.

### 2.7. Association of Pangenome Data and Epidemiological Data

The phylogenetic data generated during the pangenome analysis using Roary were combined with epidemiological information obtained from the surveillance sample collection. The resulting tree was then utilized as input for iTOL v4 [31], where visual and textual elements associated with the isolation location, host, and plague foci were incorporated into the tree.

### 2.8. Core Genome (cg) Multilocus Sequence Typing (MLST)

To perform a gene-by-gene analysis, the cgMLST method was utilized using ChewBBACA v.3.2.0 [32] which creates a genetic profiling scheme based on the core genome, followed by an allele calling process. First, the reference genome for *Y. pestis* CO92 (assembly accession GCF_000009065.1) was used to train the tool with Prodigal [24]. Next, the gene-by-gene scheme was set up using 60 complete genomes of *Y. pestis* belonging to biovars antiqua, mediaevalis, orientalis, and microtus that are available in the National Center for Biotechnology Information (NCBI) database. This step was followed by creating the whole genome (wg) Multilocus Sequence Typing (MLST) and the core genome (cg) MLST (cgMLST) using the genomes from the present study (n = 407) and the additional dataset (n = 61). Paralogs genes were excluded according to the reference genome. The minimum spanning tree algorithm (MSTree V2) implemented in Grapetree [33] was used to visualize allelic loci clusters generated by cgMLST. The complete list of the genomes used in this analysis is available in Appendix A.

## 3. Results

During the verification of the assembled genomes, it was possible to observe a homogeneous genome size and GC content in the 407 strains compatible with the known data for *Y. pestis*. The average numbers of the genomes analyzed are shown in Table 1. The data used for calculations are available in Appendix A.

Pangenome analysis resulted in a set of 5068 genes distributed among the strains. Of these genes, 3287 (64.86% of the total genome) are grouped in the core genome, while the others are distributed in the accessory genome (35.14% of the total genome). A phylogenetic tree and a matrix of the presence and absence of genes, based on this gene distribution, are shown in Figure 2. Among these clades, five main ones (C, E, F, G, and H) were found to have statistically significant genes identified through Scoary’s analysis, totaling 228 genes. These clades were highlighted for further analysis.

The resulting phylogenetic tree was integrated with epidemiological data to evaluate the prevalence of gene presence or absence based on factors such as isolation collection site (municipality or state), host (flea, rodent, or human), and plague foci (representing outbreak or endemic locations and years). However, no discernible pattern could be established between the five highlighted clades and the epidemiological data. Figure 3 displays the phylogenetic tree derived from the pangenome analysis associated with the corresponding epidemiological data.

Out of the 228 statistically relevant genes found in the five main clades, 102 were initially labeled as hypothetical proteins. Following the InterProScan analysis, we successfully reannotated 49 of these genes. However, in the case of clade F, four hypothetical proteins remained unassociated with any gene annotation. Then, we analyzed all 228 genes, searching for virulence factors using the VFDB database, which resulted in 57 genes related to virulence factors. Table 2 summarizes the distribution of putative and confirmed virulence genes and nonvirulence genes in clades C, E, F, G, and H.

Figure 4 presents the virulence-related genes enriched as absent through Scoary analysis. These absent genes are grouped by clade and organized based on their corresponding *p*-values. The complete information on all genes from all clades, including those without statistical relevance or genes not associated with virulence, along with their descriptions, annotations, and corresponding statistical values, can be found in Appendix A.

The cgMLST analysis generated new gene annotations that resulted in a set of 4088 genes among the strains, of which 3206 are in the core genome (78.42% of the total genome) and 882 genes in the accessory genome (21.58% of the total genome). The comparison of the gene distribution between the pangenomic approach using Roary and cgMLST using ChewBBACA is shown in Figure 5.

The genetic-profiling scheme generated, followed by a gene-by-gene analysis based on the core genome, produced a graph structure file using the minimum spanning tree approach which was crossed with the epidemiological data in the form of metadata in the Grapetree webservice. The resulting graph, according to the strain collection date, is represented in Figure 6. It is possible to observe three main clusters from which several evolutions depart. The first two, on the left of the image, are mainly characterized by strains from the year 1969. The third is the largest among the three main clusters, containing a greater number of strains. Most of these strains originate from the years 1974 and 1975, and are represented by two different colors. It is also possible to observe a small cluster, very characteristic of the year 1986, colored in purple derived from the main largest cluster.

An additional analysis was performed to associate the resulting graph structure with other epidemiological data. This analysis utilized the outbreak parameter, which combines the collection site and the date of positive cases of *Y. pestis*. The resulting image is represented in Figure 7. It also resulted in three main clusters with evolution points, now related to two outbreaks at Chapada do Araripe. From this first outbreak, one subcluster of strains appears self-limited to a short period of time between 1966 and 1971. But, another derived subcluster was related to an outbreak in Serra de Triunfo (colored in red). It is noteworthy that from the largest main cluster, two subclusters of strains are linked to outbreaks in Chapada da Borborema, one from 1986 colored in green (corresponding to the clusters previously colored in purple in Figure 6), and another colored in yellow (from 1979 to 1982).

## 4. Discussion

The pangenome and gene-by-gene studies showed the largest concentration of genes in the core genome, which is an expected finding since *Y. pestis* is known for its conserved genome. Even the modern strains of *Y. pestis* exhibit minimal differences from the strains of the second pandemic, particularly in terms of virulence factors [1]. The proportion of genes in the core genome was compatible between the Roary and ChewBBACA analyses. The number of genes annotated in silico in the present study is also consistent with another *Y. pestis* study [9], and even with a study that used more traditional approaches in the laboratory [34]. Another interesting finding in relation to the pangenome phylogenetic tree is the clear distinction of certain clades in the gene matrix based on groups of genes that are either present or absent in comparison to other strains.

Regarding the epidemiological features of the strains, we observed no discernible patterns between the genomic profile and the collection location, period, or host. While clade E showed a high diversity of epidemiological features, clade F showed a clearer pattern with a significant portion of the samples associated with plague foci of Araripe. The genes analyzed using the VFDB returned several virulence factors, including important well-known genes in *Y. pestis*, such as those involved in biofilm formation, iron acquisition, and cell adhesion genes [35,36,37]. The gene clusters returned by Scoary in clades C, E, G, and H are characterized by the absence of virulence genes; however, it is necessary to consider the collection period of these strains (some from the 1960s), the prolonged laboratory handling and storage process in agar stabs, in rubber-stoppered tubes, and under refrigeration at 4 °C [16].

One notable observation was the absence of genes from the *pga* family (*pgaA*, *pgaB*, *pgaC*, and *pgaD*), specifically within clade G. This gene family is renowned for its crucial role in maintaining and providing structural stability to the biofilm in various eubacteria [38] and is orthologous to the *hms* family, specific to *Y. pestis*, known for the synthesis of the extracellular matrix (ECM) in biofilm formation [36,38]. Another striking absence observed in the data from clade G was the absence of the *irp3* and *irp8* genes. This is noteworthy because genes belonging to the *irp* family play a crucial role in virulence, particularly in the synthesis of Yersiniabactin [39,40]. A study has demonstrated that the deletion of *irp8* results in the reduced virulence of the pneumonic plague [41] but the absence of these genes may be attributed to repeated subculturing, as previously demonstrated with *irp2* [42].

An unexpected absence was observed in the caf operon genes (*caf1*, *caf1A,* and *caf1M*). The *caf1* gene is considered specific for *Y. pestis*, being used even for diagnostic purposes [35], and is absent in clade H while the genes *caf1M* and *caf1A* are absent in clades H and C. This operon is important for *Y. pestis* virulence since it is responsible for the F1 antigen that inhibits the phagocytosis by macrophages [43] but other work has already demonstrated the absence of these genes, possibly due to the instability of the pFra plasmid [44]. Furthermore, it has been demonstrated that the activation of *caf1* occurs at 37 °C [43] and the gene’s function is dependent on the host and its genetic markers [45]. Another gene that is considered host dependent and, similar to *caf1*, is absent in clade C, is *psaA* [45,46]. This gene is responsible for the expression of the pH6 antigen, which plays a crucial role in the virulence of the plague by inhibiting phagocytosis by macrophages and mediating the binding of the bacillus to epithelial cells [35,46]. In addition to the *psaA*, other genes from the *psa* family (*psaE* and *psaF*) are also absent in clade C. These genes play an important regulatory role in *psaA*, mediated through distinct post-transcriptional effects under the optimal conditions of temperature and pH for *psaA* expression [47]. The absence of *caf1* and *psaA* in the strains of clade C is significant because the full pathogenicity of *Y. pestis* requires these two genes [45].

The *pld* gene, also referred to as *ymt* (Yersinia murine toxin), and highlighted as absent in clade H, is another important gene for *Y. pestis* and is considered a recent genetic adaptation that allowed transmission of the bacillus via fleas, playing an important role in their midgut survival [14,15,35]. Another important aspect observed in the results was the absence of genes associated with iron acquisition and iron transport in clades G and C. The acquisition of iron plays an important role as a virulence factor because *Y. pestis* needs to capture iron for its growth, multiplication, and biological functions [35,48,49]. Sebbane et al. (2006) [50] demonstrated the importance of iron-transport genes over iron-acquisition genes, possibly due to redundant systems in *Y. pestis*; in addition, some studies have demonstrated that reduced iron acquisition can lead to virulence attenuation [49,51].

After reannotation using InterProScan, all nine genes from clade E, which were initially annotated as hypothetical, were found to be related to the T6SS (Type VI Secretion System). The T6SS is widely distributed among Gram-negative bacteria, and functions as a delivery system for various pathogenic proteins, and *Y. pestis* is known to have multiple copies [52]. Due to limited knowledge about this secretion system in *Y. pestis*, it becomes challenging to precisely measure the extent of this impact on the virulence. Together, the absence of these virulence genes in all the main clades highlighted corroborates with previous works inferring virulence attenuation in *Y. pestis* strains from northeastern Brazil [53].

Through the gene-by-gene analysis performed by cgMLST genomic and epidemiological data were associated. Unlike the phylogenetic approach, which considers only a dichotomic relationship between one strain and another, the profile generated in the cgMLST allowed several strains to be grouped in the same node and it was also possible to infer genetic similarity based on the accumulation of allele differences. Looking at the epidemiological data, the architecture of the resulting graph associated with the year of collection (Figure 6) has three main clusters from which many evolutions departed. This is consistent given that, in Brazil, the first collections are from the year 1966 and several cases appeared until the year 1972, which are represented in two of the main clusters. After a brief epidemiological silence, several cases occurred in the years 1974 and 1975, represented by the largest main cluster. It was also observed that the reference strains, obtained from a public database and not originating from Brazil, are genetically distinct from the Brazilian collection of strains. This observation highlights that the Brazilian strains have undergone their own genetic evolution.

The resulting graph associated with outbreaks in Brazil (Figure 7) also highlighted the three main clusters, now related to the two outbreaks that occurred in the Chapada do Araripe, a large region between three northeast states of Brazil. Two of these main clusters belong to the first outbreak (1966 to 1972) and have two subclusters deriving. One shows a lineage of strains that are apparently self-limiting since almost all of them are from the municipality of Exu and only three strains are from other municipalities very close to each other. In addition, all these strains are from a short period of time (1966 and 1971). But the second derived subcluster has all the strains related to the outbreak that occurred in the municipality of Triunfo, state of Pernambuco, between 1978 and 1979 (red clusters in Figure 7). From the second outbreak in Chapada do Araripe (that began in 1973) two subclusters were derived and occurred in the Chapada da Borborema region. But, despite Chapada da Borborema encompassing five states in Northeast Brazil, all these strains are from the state of Paraíba. The first subcluster has strains from 1979 and 1982 and is from the municipality of Natuba, while the other subcluster has all strains from 1986 and is mainly in the municipality of Solânea. Together, these results enabled further understanding of the evolution of the plague in Brazil over time.

## 5. Conclusions

The pangenome and cgMLST together allowed a comprehensive analysis of Brazilian *Y. pestis* genomes available at the Fiocruz/CYP collection. The absence of several virulence genes associated with cell adhesion, iron acquisition, and biofilm formation indicates attenuation of virulence in strains from clades C, E, G, and H. The strains of clade G showed the most absence of virulence genes, including very characteristic genes of *Y. pestis* suggesting that this was the most affected group of strains. Clade E greatly benefited from the more indepth analysis of InterProScan showing the absence of genes related to T6SS, which possibly indicates an attenuation of virulence in the strains from this clade but highlights the need to further study the role of this secretion system in *Y. pestis*. Finally, clade H presented the absence of genes mostly related to plasmids and appears to be the group less affected by virulence attenuation among the clades. Through this in silico approach, it was possible to integrate genomic and epidemiological data, shedding light on evolutionary aspects and providing valuable insights into the genetic diversity and virulence of *Y. pestis* in Brazil. Moreover, the utilization of genomic information for epidemiological investigations and outbreak tracing has shown promising potential.

## Figures and Tables

**Figure 1 pathogens-12-00991-f001:**
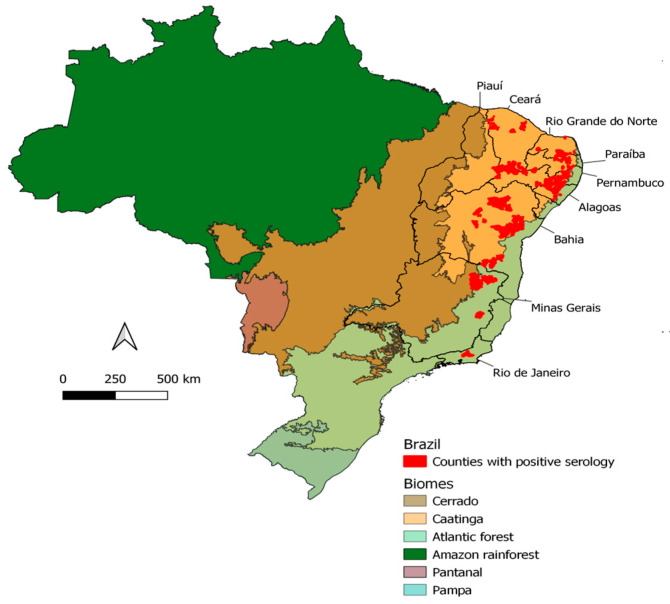
Distribution of plague cases in Brazil confirmed by bacteriological culture and serology between 1966 and 1997. Each color represents different Brazilian biomes. Red dots on the map mark the local foci where cases were registered.

**Figure 2 pathogens-12-00991-f002:**
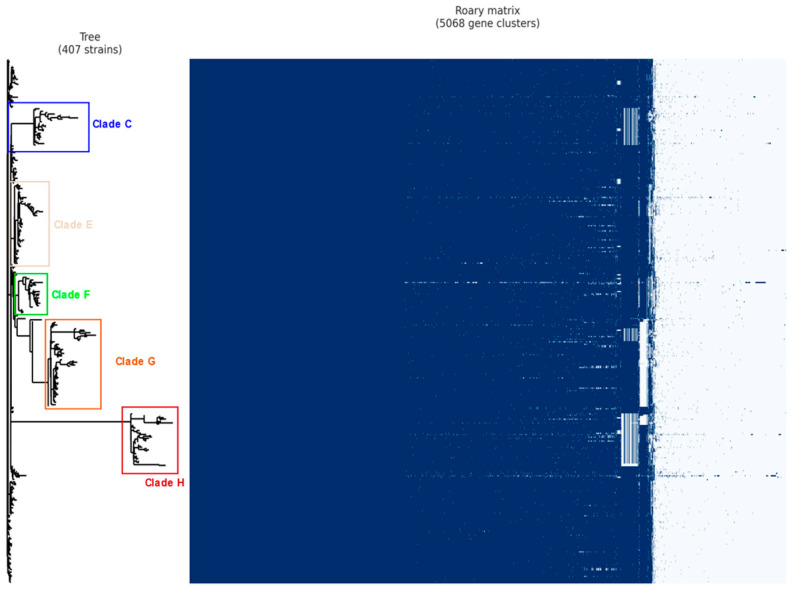
Phylogenetic tree based on the core genome and accessory genome from the pangenome analysis. In the matrix, blue marks indicate the presence of a specific gene, while white marks indicate its absence. The core genome, consisting of a dense portion of shared genes, is located at the beginning of the matrix, followed by the accessory genes in the final portion. The resulting clades are labeled from top to bottom as A to I and the five main clades: C, E, F, G, and H are highlighted in different colors.

**Figure 3 pathogens-12-00991-f003:**
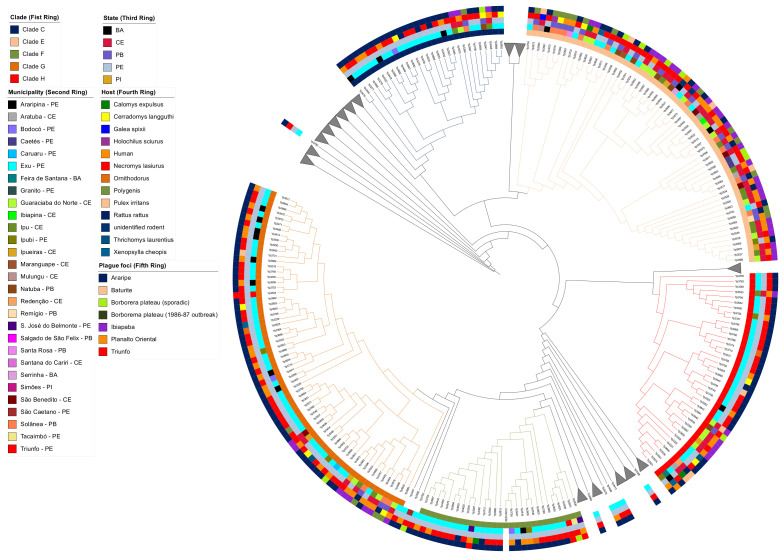
Phylogenetic tree from the pangenome analysis associated with the epidemiological data. The first ring, from the innermost to the outermost, represents the grouping of strains, with the five main clades distinguished by different colors. The subsequent rings display the association between clades and epidemiological data, including municipality, state, host, and plague foci.

**Figure 4 pathogens-12-00991-f004:**
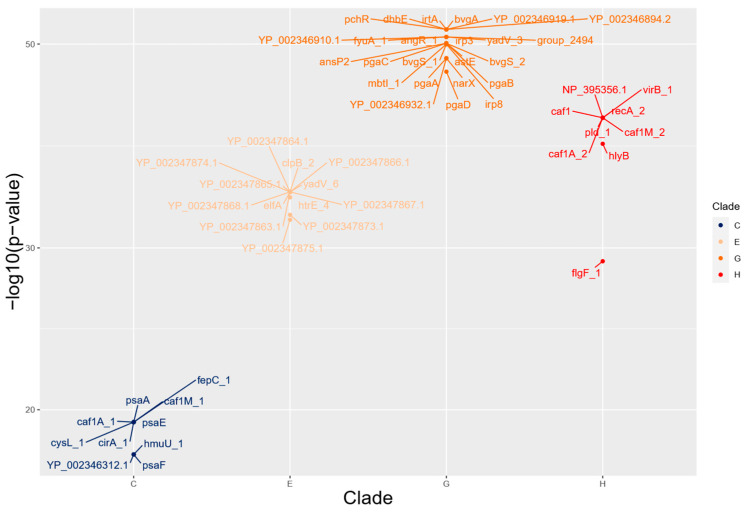
Absent virulence-related genes grouped by clade and log10(*p*-value). Each of the main clades is represented by a different color.

**Figure 5 pathogens-12-00991-f005:**
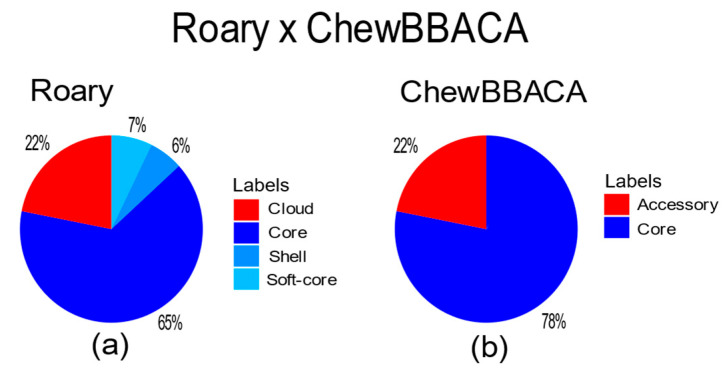
Distribution of genes after pangenome and cgMLST analysis. (**a**) shows that the largest portion of genes is in the core genome with 65% (3287 genes). The remaining genes are further classified within the accessory genome comprising 7% (386 ≤ strains ≤ 402) into soft core, 6% (61 ≤ strains ≤ 386) into shell, and 22% into cloud (strains ≤ 61). (**b**) shows the largest portion of genes in the core genome with 78% (3206 genes).

**Figure 6 pathogens-12-00991-f006:**
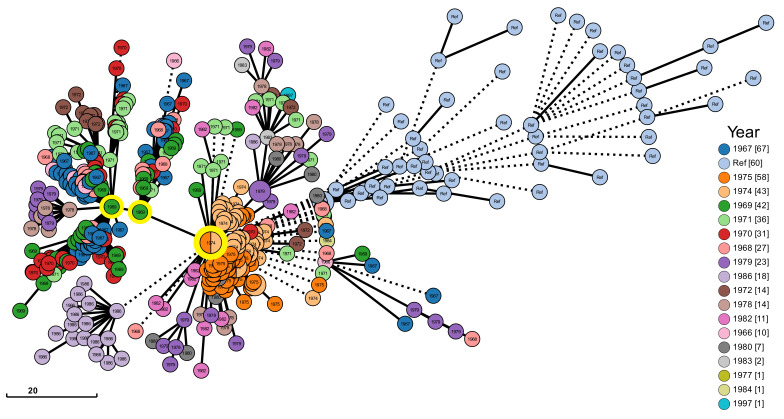
Graph showing clusters of strains based on gene-by-gene analysis. Each colored node represents a cluster of strains grouped according to allele similarity associated with the year of collection that best represents the cluster. Genetic similarity is represented by edge sizes. Three main clusters of strains representing evolutions departing are highlighted with a yellow circle. The number in brackets next to the year indicates the number of strains corresponding to the specific year. The size of each node is proportional to the number of strains clustered.

**Figure 7 pathogens-12-00991-f007:**
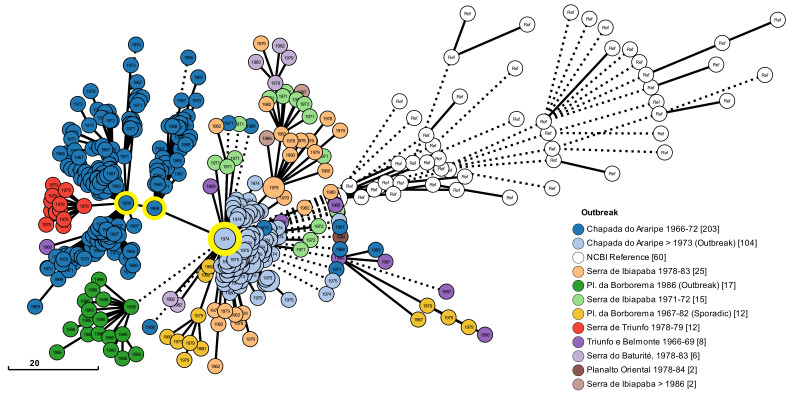
Graph showing clusters of strains based on gene-by-gene analysis. Each colored node represents a cluster of strains grouped according to allele similarity associated with *Y. pestis* outbreaks in Brazil and is labeled with the year of collection that best represents the cluster. Genetic similarity is represented by edges sizes. Three main clusters of strains representing evolutions departing are highlighted with a yellow circle. The number in brackets next to the outbreak name indicates the number of strains corresponding to the specific outbreak. The size of each node is proportional to the number of strains clustered.

**Table 1 pathogens-12-00991-t001:** General genome information.

Feature	Value
Average number of readings	2183.187
Average genome size	4.6 Mb
Average N50	367,978
Average depth	42×
GC Content	47%

**Table 2 pathogens-12-00991-t002:** Distribution of virulence and non-virulence genes in clades C, E, F, G, and H.

Clade	Putative/Virulence Genes	Nonvirulence Genes
C	10	43
E	13	0
F	0	4
G	25	41
H	9	83

## Data Availability

The data presented in this study are available in the article and Appendix A.

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
