# Peer review of "Genomic Analysis of Yersinia pestis Strains from Brazil: Search for Virulence Factors and Association with Epidemiological Data"

_pathogens, 2023, doi:10.3390/pathogens12080991_

Round 1

Reviewer 1 Report

The authors report a comparative genomic analysis using 407 Y. pestis strains isolated in different parts of Brazil, collected between 1966 and 1997, and from different hosts. The analysis identified a core genome and an accessory genome, and grouped the strains into 5 different clades. Interestingly, some clades lack known but non-essential virulence factors, such as the capsule, or virulence genes important for bubonic plague, such as the Psa pilus. Unfortunately, no discernible pattern has been correlated with epidemiological data.

I have several comments and suggestions regarding the manuscript. One of the main problems is that the cited references and several statements are incorrect. The discussion is also a mixture of results and discussion. By describing information for each clade, it often repeats the same information. Furthermore, the discussion does not really focus on the results in relation to the data reported in the literature. In particular, why certain virulence genes, such as caf or psa, might be missing. It might also make sense to see the loss of flgF, since flgF is required for motility and Y. pestis is not motile. In fact, I suggest reducing the length of the discussion by summarizing the information and discussing only the main points, i.e. discussing the reason for certain genetic losses based on the literature.

More specifically:

Lines 31 & 35, and Line 37: I think it is enough to provide just one reference.

Line 35, 38 and elsewhere: “The Plague” should be “Plague”

Line 44: I suggest citing Figure 1 after the authors had mentioned the different regions.  

ines 76-77 and elsewhere: In various places in the text, it is imperative to italicize gene names as well as the name of the bacterial species.

Lines 80-82: If I am correct, it is more around 3300 B.P. and so ~5300 years ago that a strain acquire ymt (Andrades Valtuena PNAS 2022)  

Line 83: the cited references are unrelated to the statement. Please, see Hinnebusch BJ Science 2002 and Bland D PLoS Pathog. 2021. Furthermore, Y. pestis remains confined in the midgut and proventriculus but not the intestine.

Lines 87-103: it will be useful to indicate how the strains have been maintained since they were isolated, notably considering the absence of some important virulence genes. 

Line 182: I found confusing to write that microtus is a recent biovar since it is older than Orientalis.

Lines 229-230: How are the authors sure that the genes are really virulence or non-virulence genes. Would not it be more appropriate to write putative and confirmed virulence genes and non-virulent genes?

Lines 321-323: the role of Psa may be strain or host-dependent (Anisimov J Med Microbiol. 2009; Weening Infect Immun. 2011) In fact it is somewhat the same for Caf (Weening Infect Immun. 2011). Therefore, I suggest discussing this point in the discussion.

Lines 331-338: The absence of iron transporters should also be discussed. Very few iron acquisition systems appear to be important for Y. pestis (Sebbane PNAS 2006). We do not know whether this is due to system redundancy. Therefore, I suggest discussing this point in the discussion.

Lines 349-350: Please, could you confirm that cysL is rovM? The cited reference 55 (Vadyvaloo Plos one 2015) does not mention CysL. Furthermore, the cited ref 56 (Guillouard J. Bacteriol 2002) does not mention biofilm in the mansucript.  

Lines 380: it is unclear to me whether the pga genes are the hms genes. 

Lines 406-409: Ref 69 does not consider ymt but ureD.  Please, as mentionned above, read the Bland Plos Patho 2022 on ymt, and correct the text accordingly.

Lines 451-452 and lines 464-468: I disagree to consider that the “results enabled to further understand the evolution of the plague in Brazil along time.”. I would say it provides some insights into the genetic diversity of Brazilian strains, as no experimental data are given to understanding evolution here.

Figure 1 legend: It would be useful to provide the date when the foci were investigated; i.e. between 1966 and 1997 if I have understood correctly.

Figure 2: It was unclear to me why there is a part of the figure whith dark blue and light blue. May I suggest to add a line and title indicating this is the core genome and this is the accessory genome. Does the light blue should be white?

Table S3:  By reading the pie chart of the table S3, it is unclear why the authors wrote non virulent. As such, it sounds that 75% of the strains are avirulent.

Table S3:  Is it normal that “CladoI_25_03_2022_1841 results” is empty?

Reviewer 2 Report

This study analyzed the results of the whole-genome sequencing of Yersinia pestis isolates from Brazil obtained for the period of 1966-1997, a total of 407 strains. The analysis was done using the presence/absence of gene approach, as well as cgMLST. The authors used a number of computational methods for the evaluation of genomic variations, epidemiological investigations, and potential virulence attenuation. The isolates were assigned to the clades (C, E, G, and H) which generally correlated with the years of plague outbreaks in Brazil. The strength of the study is the use of a sizable number of genomes derived from the plague foci of the country to evaluate the genetic diversity and microevolution of Y. pestis. The major weakness of this work is a lack of SNP analysis, which could further improve the resolution of the analysis, particularly within the clades.       

Other points to consider are the following:

  1. This paper must have a Table (Supplemental) with all isolates listed with the information on the year of isolation, host, location, biome, etc. The clade assignment should be shown as well.
  2. Page 3, section 2.1. The maintenance of isolates in the collection for this prolonged period of time should be described in detail. Were strains lyophilized, stored at ultralow temperature, or just refrigerated in the agar stocks with periodic refreshing? There is a big concern about genomic instability during prolonged storage, which can directly affect the results of the study.
  3. Page 8, line 253. Make clear what was used for “gene-by-gene” analysis. Was it pangenome, core genome, or individual genomes?
  4. Page 11, lines 323-324. Provide a reference that the pH 6 antigen “acts as a contributing factor to the production of Yersinia outer proteins (Yops)”, as a reviewer is unaware of this function of the pH 6 antigen.  
  5. Page 11, lines 369-371. Table S3 still lists reannotated nine genes from clade E as hypothetical proteins, rather than belonging to T6SS.       

Round 2

Reviewer 2 Report

All reviewer's concerns were addressed. No other suggestions.